# Overexpression of the High-Affinity Nitrate Transporter *OsNRT2.3b* Driven by Different Promoters in Barley Improves Yield and Nutrient Uptake Balance

**DOI:** 10.3390/ijms21041320

**Published:** 2020-02-15

**Authors:** Bingbing Luo, Man Xu, Limei Zhao, Peng Xie, Yi Chen, Wendy Harwood, Guohua Xu, Xiaorong Fan, Anthony J. Miller

**Affiliations:** 1State Key Laboratory of Crop Genetics and Germplasm Enhancement, Key Laboratory of Plant Nutrition and Fertilization in Low-Middle Reaches of the Yangtze River, Ministry of Agriculture, Nanjing Agricultural University, Nanjing 210095, China; bingbingluo@ahau.edu.cn (B.L.); 2018103100@njau.edu.cn (M.X.); 2018103101@njau.edu.cn (L.Z.); 2018803186@njau.edu.cn (P.X.); ghxu@njau.edu.cn (G.X.); 2Department of Metabolic Biology, John Innes Centre, Norwich Research Park, Norwich NR4 7UH, UK; yi.chen@jic.ac.uk (Y.C.); Wendy.Harwood@jic.ac.uk (W.H.)

**Keywords:** barley, nitrate transporter, *OsNRT2.3b*, nitrate uptake, NUE, yield

## Abstract

Improving nitrogen use efficiency (NUE) is very important for crops throughout the world. Rice mainly utilizes ammonium as an N source, but it also has four *NRT2* genes involved in nitrate transport. The *OsNRT2.3b* transporter is important for maintaining cellular pH under mixed N supplies. Overexpression of this transporter driven by a ubiquitin promoter in rice greatly improved yield and NUE. This strategy for improving the NUE of crops may also be important for other cereals such as wheat and barley, which also face the challenges of nutrient uptake balance. To test this idea, we constructed transgenic barley lines overexpressing *OsNRT2.3b*. These transgenic barley lines overexpressing the rice transporter exhibited improved growth, yield, and NUE. We demonstrated that *NRT2* family members and the partner protein *HvNAR2.3* were also up-regulated by nitrate treatment (0.2 mM) in the transgenic lines. This suggests that the expression of *OsNRT2.3b* and other *HvNRT2* family members were all up-regulated in the transgenic barley to increase the efficiency of N uptake and usage. We also compared the ubiquitin (Ubi) and a phloem-specific (RSs1) promoter-driven expression of *OsNRT2.3b*. The Ubi promoter failed to improve nutrient uptake balance, whereas the RSs1 promoter succeed in increasing the N, P, and Fe uptake balance. The nutrient uptake enhancement did not include Mn and Mg. Surprisingly, we found that the choice of promoter influenced the barley phenotype, not only increasing NUE and grain yield, but also improving nutrient uptake balance.

## 1. Introduction

Plants take up and use various N forms from soil, mostly the inorganic ions ammonium (NH_4_^+^) and nitrate (NO_3_^−^) [1,2,3]. Plant growth and development needs the presence of NO_3_^−^ [4]. To adjust to the different concentrations of NO_3_^−^ in the soil, plants have developed multiple nitrate uptake systems. There are three known nitrate uptake systems: a constitutively high-affinity transport system (cHATS); a NO_3_^−^ inducible high-affinity transport system (iHATS); and a low-affinity constitutively expressed transport system (LATS) [5,6,7,8]. The energy for the activity of these transport systems is mainly derived from the proton electrochemical gradient across membranes [9,10]. A better understanding of how plants respond to nitrate availability in soil and regulate uptake is important for increasing nitrogen use efficiency (NUE).

Additionally, cellular pH homeostasis also plays an important role in plant growth and development. The “loosening” of cell walls occurs at low pH, which leads to cell elongation [11] and growth. Normally, the amount and ratio of the two inorganic N forms can affect plant pH homeostasis in the local environment and inside cells. Within the plant, the phloem is a vascular network tissue connecting the shoot and root for transporting nutrients and communicating signals [12]. Simultaneously, phloem pH homeostasis maintains the physiological balance of the whole plant for better transporting and signaling functions in this tissue.

There are at least 53 *NRT1* and seven *NRT2* membrane nitrate transporters in Arabidopsis [9,13]. The *NRT1* family mostly includes low-affinity nitrate transporters, with the exception of *CHLI* (*AtNRT1.1*), which is a dual-affinity nitrate transporter and transceptor [14,15,16]. *CHL1* was the first nitrate transporter identified in plants [17] and may function as a nitrate sensor operating over a wide range of concentrations and having a specific involvement in nascent organ development [14,18]. *NRT2s* are high-affinity nitrate transporters that need a partner protein, NAR2, to perform their functions [19,20,21,22]. Several transporters are mainly involved in long-distance xylem and phloem nitrate transport within the plant [23,24,25], and studies have identified five *NRT2* genes in rice [26,27]. *OsNRT2.1*, *OsNRT2.2,* and *OsNRT2.3a* need to interact with the partner protein *OsNAR2.1* for nitrate uptake, whereas *OsNRT2.3b* and *OsNRT2.4* can function independently without *OsNAR2.1* [27,28,29,30]. *OsNRT2.3* has two transcripts, *OsNRT2.3a* (AK109776) and *OsNRT2.3b* (AK072215), which exhibit 94% similarity in amino acid sequences [25]. *OsNRT2.3b* is mainly expressed in the phloem and has a regulatory motif on the cytosolic side that switches nitrate transport activity on/off by a pH-sensing mechanism [10]. *OsNRT2.3b* overexpression in rice can improve the pH-buffering capacity and increase N, Fe, and P uptake [25].

Barley (*Hordeum vulgare* L.) was domesticated in the Fertile Crescent about 10,000 years ago [31]. Today, barley is an economically important crop that is mainly used in the production of malt, food, animal feed, and medicine [32]. In addition, barley is also an important model species for *Triticeae* genomics, and it has extensive physiological information on N uptake and transport [33]. Crop yield is affected by a variety of factors, of which N is a key factor. Plants usually assimilate N from nitrate (NO_3_^−^) or ammonium (NH_4_^+^) from soils [3]. However, these chemical species can also be converted to N_2_O by microbial metabolism, leading to environmental N losses and deficiency [34], which can be a serious threat to crop development and growth. 

A larger number of candidate transporter genes that may be responsible for nitrate uptake and transport have been identified in many species. In this study, we transferred the high-affinity nitrate transporter rice gene *OsNRT2.3b* into barley by using Agrobacterium-mediated transformation technology [35]. Then, we studied nutrient uptake and crop yield in the different transgenic lines. 

## 2. Results

### 2.1. Confirmation of Transgenic Barley Lines with Real-Time Quantitative PCR and Western Blotting

The segregating population of T0-generation transgenic barley transformed with *OsNRT2.3b* was selected with 50 mg/L hygromycin, and 200 lines were acquired. Then, T1 transgenic plants were screened from null segregates by the germination of seed on hygromycin (100 mg/L), and we obtained three independent T2 lines for each type of transgenic barley. The T2 generation of Ubi-1/2/3 and RSs1-1/2/3 was further characterized by real-time PCR and Western blotting. RNA and protein were extracted from stems of all barley lines in the tillering stage, and RT-PCR analyses indicated an absence of *OsNRT2.3b* transcription in the wild-type (WT) barley. However, the stems of transgenic barley lines had different expression levels of *OsNRT2.3b* mRNA (Figure 1A). The overexpression of *OsNRT2.3b* was higher in Ubi-1/2/3 transgenic lines than in RSs1-1/2/3 transgenic barley (Figure 1A). Likewise, Western blotting showed that the *OsNRT2.3b* protein just appeared in overexpression barley lines, not in wild type (Figure 1B). The expression of *OsNRT2.3b* in the Ubi-3 barley line was determined at the transcriptional and protein levels in transgenic barley lines (Figure 1). Meanwhile, the copy numbers of transgenic barley lines were identified by Southern blot (Appendix A). The results show that Ubi-1/2/3 and RSs1-1/2 were one copy insertion lines and RSs1-3 was two copy insertion line.

### 2.2. Phenotype of Transgenic Barley Lines during the Vegetative Growth Stage

Representative photographs of these transgenic barley lines during the vegetative growth stage are shown in Figure 2A,B. The Ubi-1 and Ubi-3 lines grew better than WT at the seeding stage (Figure 2A). Correspondingly, the fresh weights of Ubi-1/3 were higher than the WT (Figure 2C). By contrast, the RSs1-1/2/3 line had a better phenotype than the wild type in plant weight and tiller at the tillering stage (Figure 2B). The fresh weights of the transgenic lines were also increased at the tillering stage, besides the RSs1-3 line not being marked (Figure 2D). These results indicated that the rice nitrate transporter gene *NRT2.3b* can function in barley to increase growth. Moreover, the effect of the ubiquitin promoter driving strong gene expression in rapidly dividing cells is faster in the phenotype than in RSs1, which is a phloem-specific promoter (Figure 2A,B), or expressing *OsNRT2.3b* in the phloem of rice by in situ experiment [25,26].

### 2.3. Characterization of Transgenic Barley during the Reproductive Growth Stage

The phenotypes of all the barley lines were not significant different at the grain-filling stage. Grain-related indicators are the basic criteria for measuring crop yields. Thus, we measured the grain weight, N, and metal concentration in the seeds of all the barley lines. We found that the grain weight of the transgenic barley lines was increased relative to WT (Figure 3A). Moreover, the Ubi-1/2/3 and RSs1-1/2/3 lines were approximately 60% and 40% higher than those of the WT (Figure 3A). NUE in the transformants was higher than WT (ranging from 27% to 43%) (Figure 3C). The total N content of Ubi-1/2/3 was increased by 33%, whereas the total N of RSs1-1/2/3 lines was only increased 19% and exhibited no statistically significant difference compared to WT (Figure 3B). Furthermore, the Fe concentration in seeds was only increased in the Ubi-1/2/3 lines (Figure 3D). Additionally, we analyzed other elements in the seeds of different lines. The Ubi-1/2/3 lines exhibited no significant differences in manganese (Appendix A) and magnesium (Appendix A) concentrations relative to WT. This result was similar to the overexpression of *OsNRT2.3b* with Ubi promoter in rice seeds [36], but the manganese concentrations in RSs1-1/2/3 seeds were significantly lower than those in the WT (Appendix A). 

Seed size is an important agronomic trait; thus, we assessed seed morphology. The seeds of transgenic barley were larger than those of WT (Figure 4A, B). The seed lengths of Ubi-1/2/3 and RSs1-1/2/3 lines were increased by 9% and 10%, respectively (Figure 4A), and the seed widths of the Ubi-1/2/3 and RSs1-1/2/3 lines were increased by 24% and 27% compared with that of the WT (Figure 4B).

These results indicate that a high expression of *OsNRT2.3b* with a pH-sensing motif in barley enhances the seed length and width to increase grain weight. Meanwhile, the ubiquitin promoter-driven expression of *OsNRT2.3b* in barley resulted in an improved Fe concentration in seeds when compared with the phloem-specific expression of *OsNRT2.3b.*

### 2.4. Plant Development and Metabolism at Maturity

Physiological indexes are used to measure plant development and metabolism, and plant shoot growth is the basis for normal plant ontogenesis and crop yield. The transgenic lines exhibited an increased tiller number per plant (Figure 5A) as well as increased shoot dry weight (Figure 5B). The Ubi-1/2/3 line exhibited an increased nitrate concentration in shoot by 23%, and the RSs1-1/2/3 lines showed an increasing trend, but it was not statistically significant compared with WT (Figure 5C). The stems of transgenic lines show a trend of increased total N content but it was not significantly different (Figure 5D), which was contrary to the increased total N of seeds in transgenic lines (Figure 3B).

Taken together, *OsNRT2.3b* as a rice nitrate transporter can improve nitrate acquisition and transport in barley, which is a favored nitrate crop. This result is consistent with the overexpression of *OsNRT2.3b* in rice. [25].

### 2.5. NH_4_^+^ and NO_3_^−^ Influx Rates in WT and OsNRT2.3b Transgenic Barley

The influence of *OsNRT2.3b* overexpression on the ^15^N-NH_4_^+^ and ^15^N-NO_3_^−^ influx of hydroponically grown barley from root into intact plant was determined. Under 0.2 mM ^15^NH_4_^+^ supply, the transgenic barley lines exhibited no significant differences compared to WT (Figure 6A). In contrast, the uptake of ^15^N-NO_3_^−^ was higher influx in transgenic lines after five minutes. The nitrate influx rates of the Ubi-1/2/3 and RSs1-1/2/3 lines were higher than that of the WT (ranging from 38% to 53%) (Figure 6B). These results indicated that the overexpression of the high-affinity nitrate transporter *OsNRT2.3b* improved NO_3_^−^ uptake (Figure 6B) into the plant (Figure 5C), which resulted in an increased total N content in seeds (Figure 3B) to further enhance the grain yield.

### 2.6. The Effect of Different N Treatments on Gene Expression, and Total N, P, and Other Fe Concentration in Different Plant Parts

We investigated the effect of *OsNRT2.3b* overexpression on barley growth under different N treatments (0.2 mM NH_4_^+^ and 0.2 mM NO_3_^−^). NH_4_^+^ and NO_3_^−^ treatments up-regulated the expression of *OsNRT2.3b*, which was mainly expressed in leaves and stem sheath (Figure 7A,D). The expression of *OsNRT2.3b* in leaves of the Ubi-1/2/3 lines was increased 1.6-fold and 3.8-fold, compared to the RSs1-1/2/3 lines treated with 0.2 mM NH_4_^+^ and 0.2 mM NO_3_^−^, respectively (Figure 7A,D). Interestingly, the expression of *OsNRT2.3b* in all lines treated with 0.2 mM NO_3_^−^ was increased by more than 10-fold compared with 0.2 mM NH_4_^+^ treatment (Figure 7A,D). NH_4_^+^ treatment did not affect the dry weight of the different plant parts (Figure 7B) or the total N concentration in any of the lines (Figure 7C). In contrast, NO_3_^−^ treatment increased the dry weight of leaves and sheaths in the transgenic lines (especially the Ubi-1/2/3 line), whereas the roots remained unaffected (Figure 7E). NO_3_^−^ treatment exhibited a similar effect on the total N concentration (Figure 7F). These effects of NO_3_^−^ treatment on dry weight and total N concentration corresponded with the expression of *OsNRT2.3b* under 0.2 mM NO_3_^−^ treatment (Figure 7D).

The effects of N treatment on the expression of different nitrate transporters expression in the different plant parts were also assessed. The nitrate transporters *HvNRT2.1/2.2/2.3* and partner protein gene *HvNAR2.3* in leaves and sheaths were also up-regulated in the transgenic lines under NO_3_^−^ treatment, especially in the Ubi-1/2/3 lines (Appendix A). The barley NRT2 gene family requires a partner protein gene *HvNAR2.3* to transport nitrate [37]. Interestingly, *HvNRT2.1/2.2/2.3* and *HvNAR2.3* were not up-regulated in barley roots (Appendix A). The elemental concentrations tested in different parts of all the barley lines showed no significant differences with the 0.2 mM NH_4_^+^ and NO_3_^−^ treatments (Appendix A). From statistical analysis, NH_4_^+^ treatment did not affect the total N, P, or Fe in shoots and roots of any barley line (Appendix A). However, the shoots of transgenic barley lines exhibited a higher total N than WT under 0.2 mM NO_3_^−^ treatment. On the other hand, RSs1-1/2/3 exhibited the highest total P and Fe in shoots (Table 1). In contrast, the amounts of total N, P, and Fe exhibited an opposite trend in roots (Table 1). These results suggest that more nutrients were transported to the shoots from the roots for plant growth in the transgenic lines.

In further hydroponic experiments, barley was grown in 10 mM NH_4_^+^/NO_3_^−^. A 10 mM NH_4_^+^ and NO_3_^−^ supply resulted in an increased expression of *OsNRT2.3b* in the leaves of transgenic lines and a suppressed expression in sheaths and roots (Appendix A). The dry weights of the different plant parts (Appendix A) and the total N concentration (Appendix A) were not affected by 10 mM NH_4_^+^ and NO_3_^−^. These results indicate that high N concentrations suppress the expression of the high-affinity nitrate transporter *OsNRT2.3b* to decrease N uptake and transfer.

### 2.7. The Characteristics of HvNRT2.5, a Gene Homologous to OsNRT2.3b

Barley has seven candidate members of the *NRT2* family (*HvNRT2.1-2.7*) and three partner proteins HvNAR2 [38]. Interestingly, *OsNRT2.3b* has a high sequence similarity to *HvNRT2.5* in barley (Appendix A). Furthermore, *HvNRT2.5* has a pH-sensing motif similar to that identified in *OsNRT2.3b* [25] (Appendix A). The relative expression of *HvNRT2.5* in transgenic barley roots was up-regulated under 0.2 mM NO_3_^−^ condition, compared with the wild types (Figure 8A). However, the relative expression of *HvNRT2.5* in leaves and sheaths was not significantly different in all barley lines under 0.2 mM NO_3_^−^ supply (Figure 8A). *HvNRT2.5* transcript was clearly up-regulated in the roots under NO_3_^−^ supply (Figure 8B). Simultaneously, we found that the *HvNRT2.5* needs a partner protein *HvNAR2.3* to transport nitrate under 0.25 mM NO_3_^−^, pH 5.5 supply when expressed in oocytes (Figure 9A). The co-injection of *HvNRT2.5* and *HvNAR2.3* in oocytes did not significantly increase nitrate transport relative to water-injected controls under 10 mM NO_3_^−^, pH 5.5 treatment (Figure 9B). These data indicated that *HvNRT2.5* belongs to a high-affinity nitrate transporter and it needs a partner protein *HvNAR2.3* to transfer nitrate in oocytes.

These data suggest that a low concentration of nitrate can increase *OsNRT2.3b* gene expression in barley more than a low ammonium treatment (Appendix A, see Figure 2 in [36]), which in turn increases the transport of nutrients, and ultimately increases the grain yield in the transgenics. This may also be due to the up-regulation of *HvNRT2.5* expression in the transgenic barley root, while increasing the nitrate uptake and overexpression of *OsNRT2.3b* in transgenic lines shoot further facilitates nitrate transport.

## 3. Discussion

Plant uptake of NO_3_^−^ is carefully regulated depending on the N supply forms and N status of the plant [39]. In rice, it has been reported that the *OsNRT2* gene family has key functions in the uptake of NO_3_^−^ from soil and transport from root to shoot [10,25,26,40,41]. Normally, the *OsNRT2* family (except *OsNRT2.3b*) requires the partner protein *OsNAR2.1* for transporting nitrate [30]. *OsNRT2.3b* is one of the two transcripts with variation in expression and contains a pH-sensing motif that can regulate the transport activity under various N supply forms and improve NUE and grain yield [10]. However, it is not known whether the *OsNRT2.3b* transporter can exert a nitrate transport function in barley. In this study, the main objective was to demonstrate whether the overexpression of *OsNRT2.3b* driven by the strong promoter (ubiquitin) and the phloem-specific promoter (RSs1) in barley can enhance the transport of nutrients as well as improve NUE and grain yield.

We found that the increased biomass phenotype appeared earlier in the strong promoter (ubiquitin) *OsNRT2.3b* lines when compared with the phloem-specific promoter (RSs1) barley lines (Figure 2). The fresh weights of Ubi and RSs1 barley lines were higher than those of the WT (ranging from 28% to 77%) (Figure 2C,D). Correspondingly, the expression of *OsNRT2.3b* at the RNA and protein level in transgenic ubiquitin-driven lines was higher than that in the phloem-specific promoter lines (Figure 1). The short-term ^15^N-NO_3_^−^ influx rate was increased in the lines with an overexpression of *OsNRT2.3b*, showing that increased levels of the rice transporter in barley can increase nitrate uptake compared with WT under an external 0.2 mM NO_3_^−^ supply (Figure 6). More nitrate was transferred from the root to the shoot in the transgenic barley. Therefore, the measured nitrate concentrations in shoots in transgenic barley were higher than those in WT (Figure 5C). These data demonstrated that *OsNRT2.3b* expressed in barley is directly involved in nitrate uptake and transport, which also leads to change in phenotypes, seed morphology, and biomass in transgenic lines (Figure 2, Figure 3, Figure 4). However, the phenotypes of transgenic lines exhibited no visible significant differences during the grain-filling period (Figure 3A). One explanation is that when a critical plant size is established, the overexpression of *OsNRT2.3b* no longer influenced plant biomass, emphasizing the importance of N supply for establishing the vegetative phase of growth. Another explanation is that the phloem-specific overexpression of *OsNRT2.3b* just enriched sieve-tube nitrate, and the optimizing pH homeostasis function in phloem cells could increase the long-distance transport of total P and Fe to accumulate more biomass and grain yield during the later growth stages (Table 1). The N content in the two transgenic lines is similar, and there was a comparable P/Fe content change (Table 1). As for the reason why N content was similar in two types of transgenic plants, we think that even though the Ubi promoter was driving the expression in all the types of cells, the phloem-specific expression contributed a stronger functional influence on N transport in the plants. Indeed, the overexpression of *OsNRT2.3b* in rice can maintain a relatively low phloem sap pH, which enhanced the accumulation of total P and Fe in leaves [25]. Therefore, the overexpression of *OsNRT2.3b* driven by the phloem (RSs1) promoter can achieve a similar phenotypic effect to that provided by the strong expression of the Ubi promoter. Nevertheless, the detailed molecular mechanism remains to be revealed.

The *OsNRT2.3* gene had two transcripts: *OsNRT2.3a* and *b.* In rice, the expression of *OsNRT2.3a* occurred in roots and was induced by nitrate treatment, whereas the expression of *OsNRT2.3b* was relatively weak in roots, but it was abundant in leaves [25,42]. The expression of *OsNRT2.3b* is generally very low in WT rice [25]. Some *HvNRT2* family members require the partner protein *HvNAR2.3* to transport nitrate in barley, but not *HvNAR2.1* [37]. Some papers have reported that the *HvNRT2.1*, *HvNRT2.2,* and *HvNRT2.3* genes belong to the high-affinity nitrate transporter family [43,44]. The oocyte data shows that *HvNRT2.5* also belongs to the high-affinity nitrate transporter family (Figure 9A,B).

Interestingly, the expression pattern of *OsNRT2.3b* was affected by the external N supply forms (Figure 7A,D). One explanation is that a lower expression of *OsNRT2.3b* in NH_4_^+^ plants compared to NO_3_^−^ plants, as shown in Figure 7, may result from another level of regulation by a micro RNA [45]. Additionally, the expression of *OsNRT2.3b* was increased in Ubi transgenic barley more than in the RSs1 transgenic barley lines under 0.2 mM NO_3_^−^ supply (Appendix A). The high-affinity nitrate transporter *OsNRT2.3b* was strongly expressed in the leaves of transgenic barley under high NH_4_^+^/NO_3_^−^ supply and weakly expression in the sheath and root (Appendix A) [46]. This may be due to feedback regulation from the external N form and concentration. The expression of other HvNRT2s in transgenic lines was altered only by 0.2 mM NO_3_^−^ supply, but not NH_4_^+^ (Appendix A and Figure 8). The accumulation of *HvNRT2.1/2.2/2.3* transcript was observed in the wild-type barley roots under NO_3_^−^ supply [38]. However, the expression of *HvNRT2.1/2.2/2.3* was suppressed in the roots of the transgenic lines (Appendix A). The suppression of these genes may be due to a greater accumulation of NO_3_^−^ in the roots, which directly down-regulated the expression of *HvNRT2.1/2.2/2.3*. This was in contrast to the high-affinity nitrate transporter *HvNRT2.5* expression pattern in transgenic barley roots under 0.2 mM NO_3_^−^ supply (Figure 8). We also found that the expression of *HvNRT2.1/2.2* and partner protein gene *HvNAR2.3* were all up-regulated in the transgenic leaves to response to increased nitrate in the barley plant under 0.2 mM NO_3_^−^ supply (Appendix A). Nitrate is the predominant form of N supply for barley, and thus we suggest that NO_3_^−^ uptake through increased *OsNRT2.3b* expression leads to the induction of *HvNAR2.3* expression. These results indicate that nitrate feedback signaling in barley can alter the expression patterns of *HvNRT2s* in the transgenic lines.

The up-regulation of *HvNRT2.5* in the transgenic lines roots improved nitrate uptake, and the overexpression of *OsNRT2.3b* promoted more nitrate transport to the shoots in the transgenic lines. Transgenic technology has been successfully used to up-regulate the expression of *OsNRT2.3b* in barley. The characterization of these transgenic barley lines has demonstrated that this rice high-affinity nitrate transporter has a specific role in nitrate uptake and transport from root to shoot in barley.

## 4. Materials and Methods

### 4.1. Plant Materials

We amplified the *OsNRT2.3b* (AK072215) open reading frame (ORF) from cDNA, which was isolated from *Oryza sativa* L.ssp. Japonica cv. Nipponbare using the primers listed in Appendix A. Then, *OsNRT2.3b* was inserted into the pG3 vector by the *AscI* and *KpnI* enzyme sites without GUS(β-glucuronidase), which was driven by the RSs1 promoter [47] and transferred into the plant expression vector pB211. *OsNRT2.3b* was also cloned into the pMD19-T vector (Takara Biotechnology company, Dalian City, Liaoning Province, China) and expression vector pTCK303 with a ubiquitin promoter. The positive vector was identified by restriction digest and DNA sequencing. Next, the binary vectors pUbiquitin-*OsNRT2.3b* and pRSs1-*OsNRT2.3b* were introduced into the *Agrobacterium tumefaciens* strain AGL1, which was used to transform the immature embryos of the spring barley (*H. vulgare L.*) cultivar ‘Golden Promise’ by the method of Bartlett et al. 2008 [48] and Harwood et al., 2009 [49]. The T0-generation transgenic plants were identified by selection on 50 mg/L hygromycin. Transgenic T1 plants could be identified from null segregates by germination of the seed on hygromycin (100 mg/L) containing agar. Three independent T2 generation transgenic barley lines containing each construct were used for further analysis.

### 4.2. Growth Conditions

Barley plants (*Hordeum vulgare*
*L.* cv Golden Promise) after hygromycin selection were planted in a greenhouse with artificial illumination with 16/8 h light/darkness at 23 °C and 18 °C, respectively. Light had a photon flux density at plant level equal to 300 μmol m^−2^ s^−1^ and 60% relative humidity. For pot experiments, 10 seeds of each transgenic line were grown in pots with barley mix compost, which included perlite and grit [48].

In the hydroponic experiments, healthy barley seeds were selected, surface sterilized with 30% NaClO for 30 min, rinsed with distilled water 3 times, and then soaked for 6 h at 25 °C. Then, the seeds were germinated on moistened filter papers in dark germination boxes and placed into a growth chamber (23/18 °C, day and night). Uniformly sized 10-day-old seedlings of all the transgenic barley lines were transplanted into plastic pots for hydroponic culture in greenhouse. These seedlings were grown in 1/4 Hoagland’s nutrient solution, which was applied in the following proportions: 2 mM KNO_3_, 1 mM NH_4_NO_3_, 0.5 mM MgSO_4_, 0.5 mM CaNO_3_, 0.25 mM CaCl_2_·2H_2_O, 1 mM NaH_2_PO_4_·2H_2_O, 0.25 mM Fe_2_-EDTA, 11 μM MnCl_2_·4H_2_O, 46 μM H_3_BO_3_, 0.8 μM ZnSO_4_·7H_2_O, 0.32 μM CuSO_4_·5H_2_O, and 0.08 μM (NH_4_)_6_Mo_7_O_24_·2H_2_O. Nutrient solutions were changed every two days, and the pH was kept at 6.0 by adding the MES(2-(N-Morpholino) ethanesulfonic acid) buffer in the nutrient solution. In addition, the barley roots were aerated in the hydroponic experiment using aquarium air pumps.

### 4.3. Western Blotting

The biosynthesis and purification of anti-OsNRT2.3b rabbit monoclonal antibody were as previously described [25,26]. All tissues that were from the stems of all transgenic barley lines were homogenized and lysed in buffer containing 1% Nonidet P-40 and the protease inhibitors. Then, the lysates were clarified by centrifugation, and the protein concentration was measured by spectrophotometry with Bradford reagent in A595. Fifty micrograms of protein was boiled in gel-loading buffer and analyzed on 10% SDS-PAGE gels. After that, the protein was transferred to polyvinylidene difluoride membranes and hybridized by first antibody *OsNRT2.3b* (1:500) and *Actin* (1:5000) overnight at 4 °C. The membrane was incubated in an appropriate second antibody (1:5000), followed by chemiluminescence detection to check the protein brands.

### 4.4. DNA/RNA Extraction and qPCR Analysis

Total RNA was extracted from 100 mg of tissue using TRIzol (Invitrogen, Carlsbad, USA). RNA (2 μg) was reverse-transcribed into cDNA using HiScript Reverse Transcriptase (Vazyme, Nanjing, China), according to the manufacturers protocol. Real-time PCR was used with a Power SYBR Green Master Mix (Vazyme, Nanjing, China) for target genes and *HvActin* [50]. Real-time PCR was performed using the gene specific primers shown in Appendix A. The PCR parameters for the detection of *HvActin* [50], *OsNRT2.3b* (AK072215), *HvNRT2.1* (accession no. U34198), *HvNRT2.2* (accession no. U34298), *HvNRT2.5* (accession no.DQ539042) and *HvNAR2.3* (accession no. AY2535450) were 95 °C for 30 s, followed by 40 cycles of 95 °C for 10 s, 60 °C for 30 s, and 72 °C for 10 s. Genomic DNA was extracted as described by Ausubel et al. (1994) [51].

### 4.5. Southern Blot Analysis

Transgene copy numbers were identified by Southern blot analysis. First, genomic DNA were extracted from the leaves of WT and transgenic barley lines. Then, DNA was digested by Hind Ш and EcoR Ι. Second, the digested DNA was separated on 1% (*w*/*v*) agarose gels, transferred to a Hybond-N^+^ nylon membrane, and hybridized using the hygromycin-resistant gene.

### 4.6. Determination of ^15^N-NH_4_^+^/NO_3_^−^ Influx Rate in Different Barley Lines

Barley seedlings of untransformed control plants or wild-type (WT) and *OsNRT2.3b* transgenic barley lines were planted in 1/4 Hoagland’s nutrient solution for 2 weeks (as described above) and then N-starved for 4 days. Then, the transgenic barley lines were transferred into 0.1 mM CaSO4 for 1 min and then into complete nutrient solution containing either 0.2 mM ^15^NH_4_^+^ or 0.2 mM ^15^NO_3_^−^ (atom% ^15^N: 99%) for 5 min and finally to 0.1 mM CaSO_4_ for 1 min [52]. The ^15^N influx rate was calculated depending on methods described by Chen et al. (2016) [46].

### 4.7. Analysis of Agronomic Traits

The agronomic traits were measured for all the transgenic barley at the seeding stage, anthesis stage, and grain-filling stage. The seed morphology of transgenic barley lines was also determined. Plant fresh weight, seed weight, tillering number, and seed length/width were measured. Detailed methods for measurement of these agronomic traits were described previously [53].

### 4.8. Measurements of Dry Weight, Nitrate, Total N, and Metal Ion Accumulation

WT untransformed and transgenic barley lines were harvested at the mature stage (*n* = 4) and dried at 105 °C for 30 min. Then, shoots were dried for 3 days at 75 °C. Barley from hydroponic experiments was divided into root, stem sheath, and leaves. Dry weight was measured as biomass. The total N accumulation was measured by using the Kjeldahl method [43] and was assessed in different plant parts by multiplying the N concentration by the corresponding biomass. The calculating method of NUE (NUE, g/g) was described in Chen et al. 2016 [46]. Dried samples were digested in concentrated HNO_3_ at 120 °C until no nitrogen oxide gas was released. These samples were further digested with HClO_4_ at 180 °C, at which point they become transparent. Then, samples were diluted with ultrapure water, and the concentration of metal elements in the digestion solutions were analyzed by using ICP-OES (Inductively Coupled Plasma Optical Emission Spectrometer) (iCAP6300).

### 4.9. Cloning and cRNA Synthesis of HvNRT2.5, HvNAR2.1, HvNAR2.2, HvNAR2.3, and AtNPF6.3 (CHL1) and Nitrate Uptake Assay in Xenopus Laevis Oocytes

*HvNRT2.5* (accession no. DQ539042), *HvNAR2.1* (accession no. AY253448), *HvNAR2.2* (accession no. AY253449), *HvNAR2.3* (accession no. AY2535450) and *CHL1* (At1G12110) constructs were as described previously [46]. The cRNA was transcribed by using an Ambion mMessage mMachine^®^T7 kit. The protocols for oocyte preparation, incubation, and injection of gene cRNA were the same as previously described by Feng et al. 2013 [54]. The measurements of ^15^NO_3_^−^ influx in oocytes were performed as described by Feng et al. (2013) [54].

### 4.10. Statistical Analysis

All data were analyzed using one-way analysis of variance (ANOVA). Different letters showed a significant difference between transgenic barley line and wild types. Statistically significantly differences at the *p* < 0.05 level (one-way ANOVA) were performed using IBM SPSS Statistics version 20 software (SPSS Inc., Chicago, IL, USA).

## 5. Conclusions

In conclusion, the present study has demonstrated that the overexpression of *OsNRT2.3b* in barley affects the transport of N, P, and Fe. Even when the promoters were different, the N/P/Fe contents were similar, showing analogous phenotypes in the transgenic plants. This finding may be due to N/P/Fe synergism in the plant, which was driven by the N use and pH balance in these transgenic plants. This synergistic relationship between nutrients is worthy of further investigation. We have demonstrated a novel mechanism to optimize the gene expression in plants in which a targeted promoter and not a strong promoter is used to drive gene expression. The overexpression of *OsNRT2.3b* driven by two types of promoter in barley affected the potential to enhance the transport of nutrients and improve NUE. Furthermore, the transgenic data suggest that *HvNRT2.5* has an important role in NUE in barley.

## Figures and Tables

**Figure 1 ijms-21-01320-f001:**
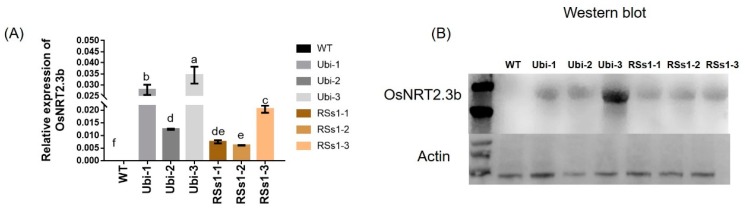
The *OsNRT2.3b* mRNA and protein expression levels in transgenic lines and WT barley plants. All plants were grown in soil in pots. (**A**) Real-time quantitative PCR analysis of *OsNRT2.3b* mRNA. Different letters indicate a significant difference between transgenic and WT plants (*p* < 0.05, one-way ANOVA). Error bars: standard error (*n* = 4 plants). (**B**) Western blot analysis of *OsNRT2.3b* expression in the barley stem. Total proteins were separated by SDS-PAGE, transferred to a polyvinylidene fluoride membrane (Thermo), and hybridized with an *OsNRT2.3b*-specific antibody. Each lane was loaded with an equal quantity of proteins (30 μg). WT: wild-type; Ubi-1/2/3 and RSs1-1/2/3 overexpression were performed with the ubiquitin promoter and phloem-specific promoter (RSs1) driven *OsNRT2.3b* gene, respectively, as below.

**Figure 2 ijms-21-01320-f002:**
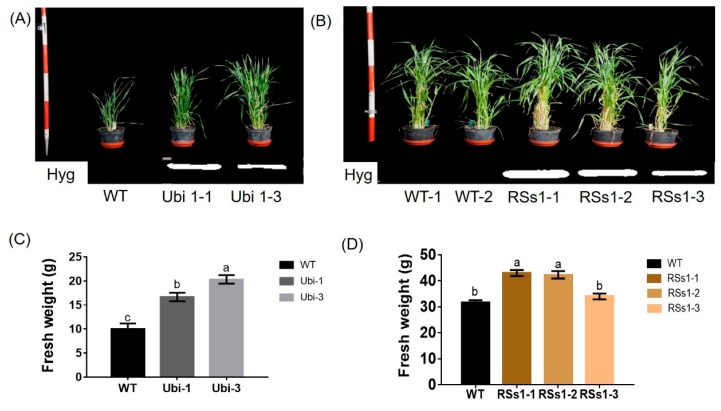
Phenotypes of all barley plants in pot experiments during the vegetative growth stage. (**A**) The phenotypes and (**C**) fresh weight of WT and Ubi-1/2/3 transgenic lines. (**B**) The phenotypes and (**D**) fresh weight of WT and RSs1-1/2/3 transgenic lines. Different letters indicate a significant difference between overexpression transgenic lines and WT (*p* < 0.05, one-way ANOVA). Error bars: standard error (*n* = 4 plants).

**Figure 3 ijms-21-01320-f003:**
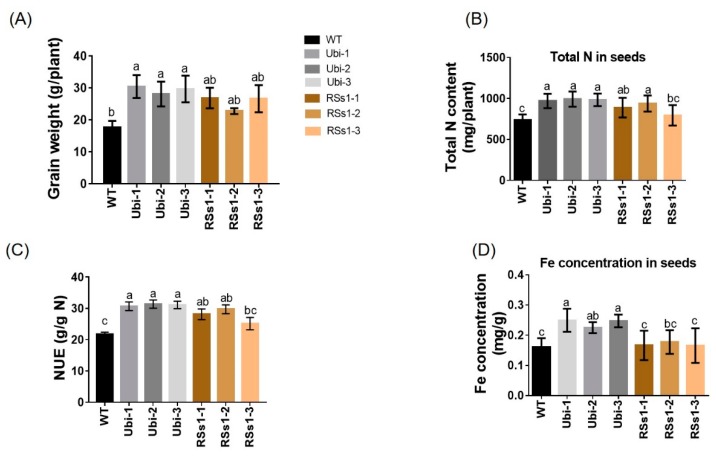
Characterization of barley plants in pot experiments at maturity. (**A**) Grain weight, (**B**) total N contents in seeds, (**C**) nitrogen use efficiency (NUE), and (**D**) Fe concentration in seeds. Different letters indicate a significant difference between transgenic lines and WT (*p* < 0.05, one-way ANOVA). Error bars: standard error (*n* = 4 plants).

**Figure 4 ijms-21-01320-f004:**
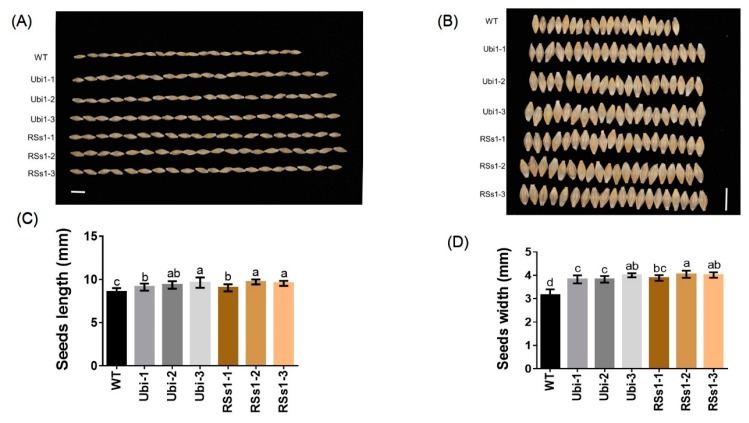
Comparison of seed morphology in barley plants (**A**), (**B**) seed morphology images of all barley plants. Bars = 1 cm. (**C**) The length and (**D**) width of seeds. Different letters indicate a significant difference between transgenic lines and WT (*p* < 0.05, one-way ANOVA). Error bars: standard error (*n* = 4 plants). Bars = 1 cm.

**Figure 5 ijms-21-01320-f005:**
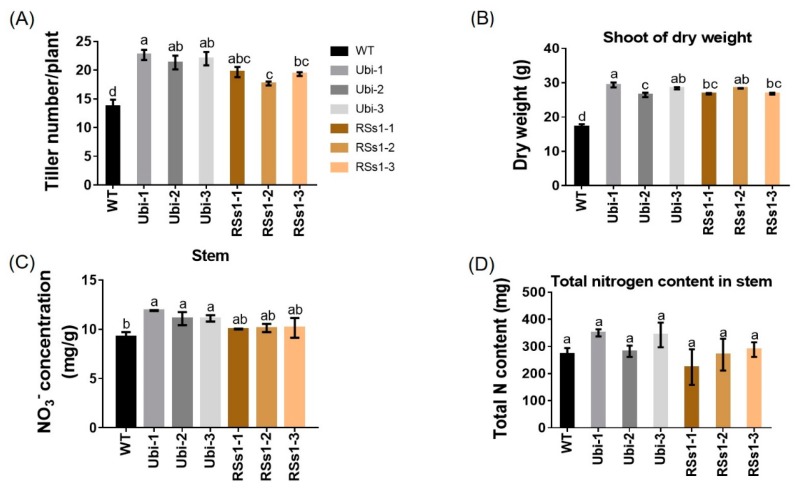
Physiological indexes of shoots. (**A**) Tiller number, (**B**) dry weight, (**C**) nitrate concentration, and (**D**) total N content in shoots at maturity in pot experiments. Different letters indicate a significant difference between transgenic lines and WT (*p* < 0.05, one-way ANOVA). Error bars: standard error (*n* = 4 plants).

**Figure 6 ijms-21-01320-f006:**
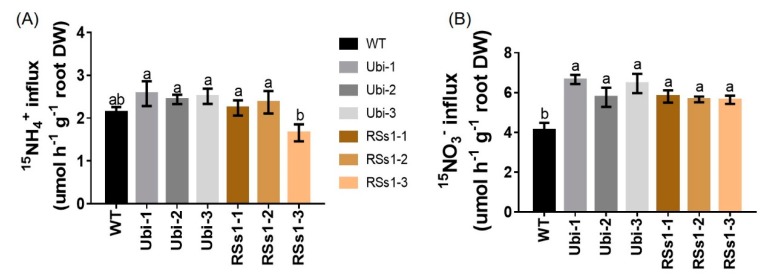
NH_4_^+^ and NO_3_^−^ influx rates of WT and transgenic plants measured using ^15^N-enriched sources. WT and transgenic barley seedlings were grown in 1/4 Hoagland nutrient solution for 2 weeks and N starved for 4 days. Then, NO_3_^−^ and NH_4_^+^ influx rates were measured at (**A**) 0.2 mM ^15^NH_4_^+^ (**B**) 0.2 mM ^15^NO_3_^−^ for 5 min. Different letters indicate a significant difference between transgenic and WT lines (*p* < 0.05, one-way ANOVA). Error bars: standard error (*n* = 4 plants).

**Figure 7 ijms-21-01320-f007:**
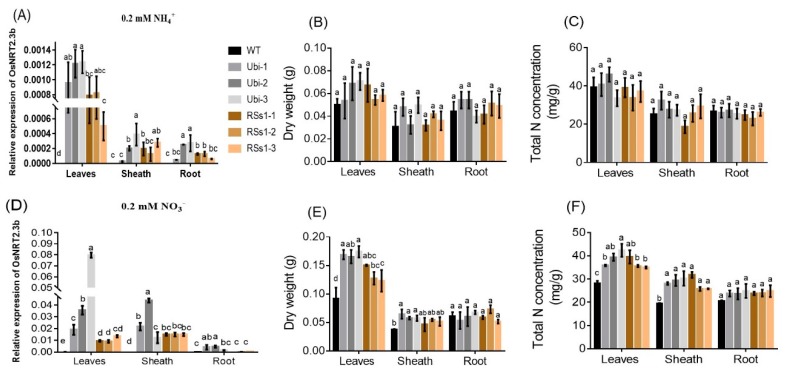
The effect of different N supplies on the expression of *OsNRT2.3b*, plant dry weight, and total N concentration in different plant parts in hydroponic experiment. (**A**–**C**) 0.2 mM NH_4_^+^ treatment, (**A**) the relative expression of *OsNRT2.3b*; (**B**) the dry weight; (**C**) the total N concentration of different parts in all barley lines. (**D**–**F**) Under 0.2 mM NO_3_^−^ treatment, (**D**) relative expression of *OsNRT2.3b*; (**E**) the dry weight; (**F**) the total nitrogen concentration of different parts in all barley lines. Error bars: standard error (*n* = 4 plants). Significant differences between transgenic and WT lines are indicated by different letters (*p* < 0.05, one-way ANOVA).

**Figure 8 ijms-21-01320-f008:**
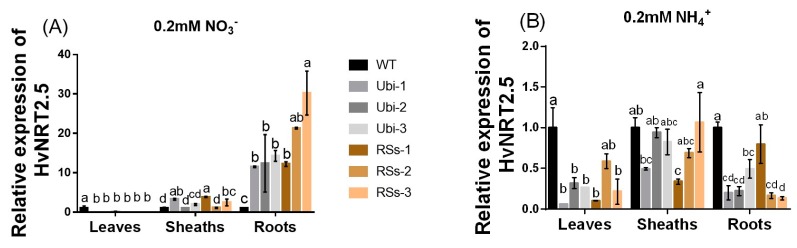
The relative expression of the *HvNRT2.5* gene homologous to *OsNRT2.3b* under 0.2 mM NH_4_^+/^NO_3_^−^ treatments. The relative expression of *HvNRT2.5* in leaves, sheaths, and roots (**A**) under 0.2 mM NO_3_^−^ supply; (**B**) under 0.2mM NH_4_^+^ condition. Error bars: standard error (*n* = 4 plants). Significant differences between transgenic and WT lines are indicated by different letters (*p* < 0.05, one-way ANOVA).

**Figure 9 ijms-21-01320-f009:**
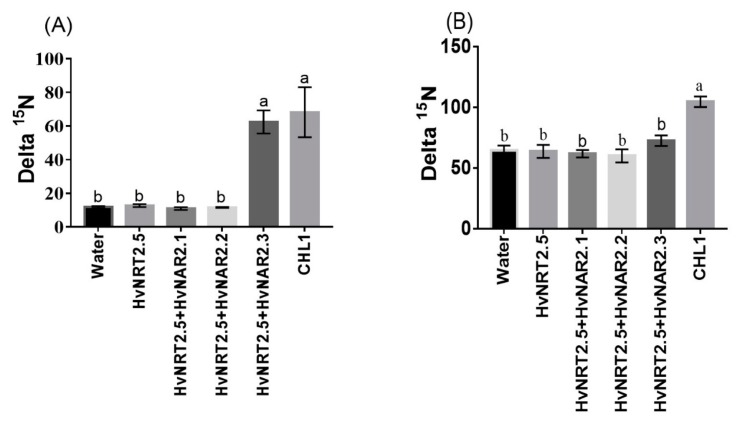
Functional assay of *HvNRT2.5* in Xenopus oocytes. Functional assay of *HvNRT2.5*, *HvNAR2.1*, *HvNAR2.2*, *HvNAR2.3*, and *CHL1* activity in nitrate uptake in Xenopus oocytes. Uptake of ^15^NO_3_^−^ into oocytes injected with water, single cRNA, or mixtures as indicated. Oocytes injected with cRNA and incubated for 10 h in modified Barth’s saline containing (**A**) 0.25 mM ^15^NO_3_^−^ at pH 5.5, (**B**) at pH 5.5; 10 mM ^15^NO_3_^−^. The values are means +/-SE for seven oocytes for every concentration. Different letters indicate significant differences between the water and cRNA-injected oocytes of the same treatments (*p* < 0.05, one-way ANOVA). Three batches of oocytes were used for each test.

**Table 1 ijms-21-01320-t001:** The effect of 0.2 mM NO_3_^−^ treatment on the distribution of total N, P, and Fe in shoots and roots. Significant differences between the transgenic and WT lines are indicated by different letters (*p* < 0.05, one-way ANOVA)

Distribution ration of shoot (%)	WT	Ubi-1	Ubi-2	Ubi-3	RSs-1	RSs-2	RSs-3
Total N	72.60b	86.76a	85.24ab	84.40ab	84.10ab	81.49ab	81.38ab
Total P	66.20b	79.60a	78.29a	75.07a	74.34a	83.15a	82.38a
Total Fe	11.49b	24.22b	23.87b	32.24b	31.48b	91.16a	80.79a
Distribution ration of root (%)	WT	Ubi-1	Ubi-2	Ubi-3	RSs-1	RSs-2	RSs-3
Total N	27.40a	13.24b	14.76ab	15.60ab	15.90ab	15.81ab	18.62ab
Total P	32.97a	20.40b	21.71b	24.93ab	25.66ab	16.85b	17.62b
Total Fe	88.51a	75.78a	76.13a	67.76a	68.52a	8.84b	19.21b

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
