# Peer review of "Overexpression of the High-Affinity Nitrate Transporter OsNRT2.3b Driven by Different Promoters in Barley Improves Yield and Nutrient Uptake Balance"

_ijms, 2020, doi:10.3390/ijms21041320_

Round 1
Reviewer 1 Report
The objectives of the current study were to demonstrate whether the overexpression of OsNRT2.3b, driven by the strong promoter (ubiquitin) and the phloem-specific promoter (RSs1) can enhance the transport of nutrients, improve NUE and grain yield in barley. The results of the current study are valuable for basic knowledge but also for plant breeding. The results show an increase of grain yield along with N amount in shoots and in grains. I recommend publishing this manuscript but only after revision of the manuscript, according to the comments below.
Main comment: Barley plants were transformed using two promoters, using two transformation systems: “OsNRT2.3b inserted into the pG3 vector by the AscI and KpnI enzyme sites without GUS driven by the RSs1 promoter and transferred into the plant expression vector pB211. OsNRT2.3b was also cloned into the pMD19-T vector (Takara Biotechnology, Dalian, China) and expression vector pTCK303 with a ubiquitin promoter”. The results show that while the difference between WT and all transformants are mostly high and significant, the differences between Ubi and RSs were not always significant. Therefore, although authors can indicate that the Ubi promoter was advantageous in few parameters I suggest that the paper should NOT focus on the difference between promoters but on the effect of the overexpression of OsNRT2.3b in barley, under two types of transformation systems and promoters, and on the potential to enhance the transport of nutrients and improve NUE. The title of the paper should be modified accordingly. If the authors insist in focusing on the differences between promoters, they should be able to answer why there is a difference, considering that they used two different vectors for transformation. Authors can add results of negative controls (if they have) for each transformation system (transformation with no promoter). The discussion includes some speculations trying to explain why there was a difference, this should include relevant references.
Some of the comets below refer to specific lines but similar problems exist in other places. Additional round of a Scientific English editing is still required. The discussion should be reorganized according to the objective of the study as suggested above. The results and discussion should be carefully written using supportive references, especially the conclusion part.
Additional comments:
Please add sequences numbers which were used for the design of primers/ transformation vectors) 2: The title of the figure describes both types of transformants at the young stage. However, in the text: “By contrast, 195 the RSs1-1/2/3 line had a better phenotype than the wild type in plant weight and tiller at the tillering stage (Fig. 2B). Please correct.
In addition, the authors describe that Ubi-1 and Ubi-3 lines grew better than WT at the seeding stage, please show RSs transformed lines at this stage (Fig. 2A); the authors describe that RSs1-1/2/3 lines had a better phenotype than the wild type in plant weight and tiller at the tillering stage, please add information on the Ubi transformed plants. Instead of Images the authors can provide a table with statistical analysis showing the advantage of each system in seedling or in tillering stages.
Line 208: However, the phenotypes”…. – can not start section with however (editing). There are (too) many figures both in the body of manuscript and in supplementary material. Some of the figures can be summarized in a table, some are not informative (i.e. 3A) and can be replaced by saying that: no significant differences were found in….”. When there are interesting results authors can summarize that : the measured variables, GY, total N, NUE, there were significantly difference between WT and all transformants but not between the Ubi and Rss promoter types, while Fe was higher only in the Ubi promoter transformants. When there is no significant differences between the Ubi and Rss promoters author cannot say that: “The NUE of transgenic barley with Ubi or RSs1 promoter was increased 43% and 27%, respectively (Fig. 3D)”. Replace it with “NUE in the transformants was higher than WT (ranged from 27% to 43%, respectively (Fig. 3D). This type of description can be found in other traits/figures (e.g. line 262; Line 363- lines were increased 77% and 28% - this is misleading). Line 239: “The ubiquitin promoter driven expression of OsNRT2.3b in barley resulted in improved seed quality when compared with phloem specific expression of OsNRT2.3b” – Not correct since it improved only Fe. 5: Physiological indexes of shoots – these are not physiological indices. More correct is to use “plant development and metabolism”. Differences in Nitrate concentration – again, there is no significant different between Ubi and Rss (only between WT and transformants). A trend of increase also in total N but not significant. Line 274: “We further went on to investigate” - change to: we investigated…(editing). Line 275: modify: on the growth of barley growth…(editing). Line 320: The authors describe that “barley has seven candidate members of the NRT2 family (HvNRT2.1-2.7) and three 3 partner proteins HvNAR2”. The authors chose to further study HvNRT2 since it has high similarity. It is not clear from the text what is the similarity of this gene as compared with the rest, therefore they should describe it in more details. Figure S2 does not describe the similarity between 3b and HvNRT2.5. Line 324: “Besides, the relative expression of HvNRT2.5 in leaves, sheaths and roots in all barley lines showed irregularity under 0.2mM NH4 condition (Fig. 8B)” – need to modify since HvNRT2.5 was clearly up regulated in the roots under NO3 but down regulated in leaves and roots of all transformants as compared with WT. Line 396 – “The explanation is that a lower expression of OsNRT2.3b in NH4 + plants compared to NO3 - plants as it is shown in Fig. 7 would be regulation by a micro RNA”. - Please add reference.
Author Response
Reviewer1
Overexpression of the high-affinity nitrate transporter OsNRT2.3b by driven different promoters in barley influences the impact on yield and nutrient uptake balance
The objectives of the current study were to demonstrate whether the overexpression of OsNRT2.3b, driven by the strong promoter (ubiquitin) and the phloem-specific promoter (RSs1) can enhance the transport of nutrients, improve NUE and grain yield in barley. The results of the current study are valuable for basic knowledge but also for plant breeding. The results show an increase of grain yield along with N amount in shoots and in grains. I recommend publishing this manuscript but only after revision of the manuscript, according to the comments below.
Point 1:
Main comment: Barley plants were transformed using two promoters, using two transformation systems: “OsNRT2.3b inserted into the pG3 vector by the AscI and KpnI enzyme sites without GUS driven by the RSs1 promoter and transferred into the plant expression vector pB211. OsNRT2.3b was also cloned into the pMD19-T vector (Takara Biotechnology, Dalian, China) and expression vector pTCK303 with a ubiquitin promoter”. The results show that while the difference between WT and all transformants are mostly high and significant, the differences between Ubi and RSs were not always significant. Therefore, although authors can indicate that the Ubi promoter was advantageous in few parameters I suggest that the paper should NOT focus on the difference between promoters but on the effect of the overexpression of OsNRT2.3b in barley, under two types of transformation systems and promoters, and on the potential to enhance the transport of nutrients and improve NUE. The title of the paper should be modified accordingly. If the authors insist in focusing on the differences between promoters, they should be able to answer why there is a difference, considering that they used two different vectors for transformation. Authors can add results of negative controls (if they have) for each transformation system (transformation with no promoter). The discussion includes some speculations trying to explain why there was a difference, this should include relevant references.
Some of the comets below refer to specific lines but similar problems exist in other places. Additional round of a Scientific English editing is still required. The discussion should be reorganized according to the objective of the study as suggested above. The results and discussion should be carefully written using supportive references, especially the conclusion part.
Response 1: We have modified the title and conclusion part according to your suggestions. We focused on the effect of overexpression of OsNRT2.3b by different promoters to improve barley nutrient balance and yield. Please see the line from 2 to 5 and line from 924 to 932 in the revised version with trackers.
Additional comments:
Point 2: Please add sequences numbers which were used for the design of primers/ transformation vectors)
Response 2: We have added the sequences numbers. Pleases see the line from 891 to 892 and line from line 926 to line 927 in the revised version with trackers.
Point 3: The title of the figure describes both types of transformants at the young stage. However, in the text: “By contrast, 195 the RSs1-1/2/3 line had a better phenotype than the wild type in plant weight and tiller at the tillering stage (Fig. 2B). Please correct.
Response3: We revised the title of figure 2. Please see the line from 202 to 203 in the revised version with tracker.
Point 4: In addition, the authors describe that Ubi-1 and Ubi-3 lines grew better than WT at the seeding stage, please show RSs transformed lines at this stage (Fig. 2A); the authors describe that RSs1-1/2/3 lines had a better phenotype than the wild type in plant weight and tiller at the tillering stage, please add information on the Ubi transformed plants. Instead of Images the authors can provide a table with statistical analysis showing the advantage of each system in seedling or in tillering stages.
Response 4: Your suggestions are very good. But we just counted the barley lines with significant phenotypic differences in each period. It is proved that overexpression of OsNRT2.3b in barley by driven two types promoters were different.
Point 5: Line 208: However, the phenotypes”…. – can not start section with however (editing). There are (too) many figures both in the body of manuscript and in supplementary material. Some of the figures can be summarized in a table, some are not informative (i.e. 3A) and can be replaced by saying that: no significant differences were found in….”.
Response 5: Ok, we have deleted the picture which is not significant difference in phenotypes of all barley lines at maturity and used be replaced by a sentence. Please see the line 224 in the revised version with tracker.
Point 6: When there are interesting results authors can summarize that: the measured variables, GY, total N, NUE, there were significantly difference between WT and all transformants but not between the Ubi and Rss promoter types, while Fe was higher only in the Ubi promoter transformants. When there is no significant differences between the Ubi and Rss promoters author cannot say that: “The NUE of transgenic barley with Ubi or RSs1 promoter was increased 43% and 27%, respectively (Fig. 3D)”. Replace it with “NUE in the transformants was higher than WT (ranged from 27% to 43%, respectively (Fig. 3D). This type of description can be found in other traits/figures (e.g. line 262; Line 363- lines were increased 77% and 28% - this is misleading).
Response 6: We have modified them according to your suggestions. Thank you very much. Please see the line 228, line229, line397 in the revised version with tracker.
Point 7: Line 239: “The ubiquitin promoter driven expression of OsNRT2.3b in barley resulted in improved seed quality when compared with phloem specific expression of OsNRT2.3b” – Not correct since it improved only Fe.
Response 7: We have revised this sentence. Please see the line 363 in the revised version with tracker.
Point 8: Physiological indexes of shoots – these are not physiological indices. More correct is to use “plant development and metabolism”. Differences in Nitrate concentration – again, there is no significant different between Ubi and Rss (only between WT and transformants). A trend of increase also in total N but not significant.
Response 8: We have modified them according to your suggestions. Thank you very much. Please see the line 371 in the revised version with tracker.
Point 9: Line 274: “We further went on to investigate” - change to: we investigated…(editing).
Response 9: Thank you for your suggestion. Pleases see the line 416 in the revised version with tracker.
Point 10: Line 275: modify: on the growth of barley growth…(editing).
Response 10: We have modified this sentence. Please see the line 416 in the revised version with tracker.
Point 11: Line 320: The authors describe that “barley has seven candidate members of the NRT2 family (HvNRT2.1-2.7) and three 3 partner proteins HvNAR2”. The authors chose to further study HvNRT2 since it has high similarity. It is not clear from the text what is the similarity of this gene as compared with the rest, therefore they should describe it in more details. Figure S2 does not describe the similarity between 3b and HvNRT2.5.
Response 11: We added HvNRT2.5 and OsNRT2.3b protein sequence alignment. Please see the figure S2B. Besides, HvNRT2.5 which has the pH sensing motif that was identified in OsNRT2.3b by marking with red box. (Fan et al., 2016). Please see the line 469 to 471 in the revised version with tracker.
Point 12: Line 324: “Besides, the relative expression of HvNRT2.5 in leaves, sheaths and roots in all barley lines showed irregularity under 0.2mM NH4 condition (Fig. 8B)” – need to modify since HvNRT2.5 was clearly up regulated in the roots under NO3 but down regulated in leaves and roots of all transformants as compared with WT.
Response 12: We have revised this sentence. Please see the line from 473 to 474 in the revised version with tracker.
Point 13: Line 396 – “The explanation is that a lower expression of OsNRT2.3b in NH4 + plants compared to NO3 - plants as it is shown in Fig. 7 would be regulation by a micro RNA”. - Please add reference.
Response 13: This explanation is only a conjecture and we have added relative reference. Please see the reference 47 in the revised version with tracker.
Reviewer 2 Report
This paper is focussed on transfer the high-affinity nitrate transporter rice gene OsNRT2.3b into barley by using Agrobacterium-mediated transformation technology. The manuscript fits within the scope of the journal. The manuscript is interesting and the idea is nice. The title is clear and it is adequate to the content of the article. The study methods are explained clearly. The body of the paper is clear and does not contain major errors. The conclusions or summary are accurate and supported by the content.
I have some recommendations for authors:
- Language style and overall style of the paper should be improved
- The reference style should be checked, as follow. Line 84,85, 123, 136, 148, 158, 159, 299, 346 – include reference number
- All latin scientific names and abbreviations of plant, bacteria, and fungi must be written in italics. Please carefully check the text.
- Please highlight the degree of novelty and originality of the work.
- Include in the text potential research directions.
Author Response
Review 2
Comments and Suggestions for Authors
This paper is focused on transfer the high-affinity nitrate transporter rice gene OsNRT2.3b into barley by using Agrobacterium-mediated transformation technology. The manuscript fits within the scope of the journal. The manuscript is interesting and the idea is nice. The title is clear and it is adequate to the content of the article. The study methods are explained clearly. The body of the paper is clear and does not contain major errors. The conclusions or summary are accurate and supported by the content.
I have some recommendations for authors:
Point 1: Language style and overall style of the paper should be improved
Response 1: Thank you for your suggestions. We have modified the language style in the whole of paper. Please see the revised paper.
Point 2: - The reference style should be checked, as follow. Line 84,85, 123, 136, 148, 158, 159, 299, 346 – include reference number
Response 2: We have revised them and modified the reference style. Please see the line 816, 817, 895,907,919,931,932 in the revised version with tracker.
Point 3: - All latin scientific names and abbreviations of plant, bacteria, and fungi must be written in italics. Please carefully check the text.
Response 3: We have modified the all latin scientific names and abbreviations of plant, bacteria, and fungi in italics. Please see the MATERIALS AND METHODS part in the revised version with tracker.
Point 4: - Please highlight the degree of novelty and originality of the work.
Response 4: Your suggestions is fine. We have modified the discussion and conclusion parts in the revised paper according your suggestions.
Point 5: - Include in the text potential research direction
Response 5: Thank you for suggestion. We have rewritten and modified this paper according to your suggestions.